# Cayley Maze: Universal Open-Ended Reinforcement Learning Environment

## Abstract

Parametrizable environments with variable complexity are crucial for advancing fields such as Unsupervised Environment Design (UED), Open-Ended Learning, Curriculum Learning, and Meta Reinforcement Learning. However, the selection of environments in evaluation procedures, along with their complexities, is often either neglected or lacks formal justification. We propose the formal definition of complexity for Markov Decision Processes using Deterministic Finite Automata and Group Theory machinery. We introduce Cayley Maze, a novel open-ended reinforcement learning environment that naturally generalizes problems like solving Rubik's Cube or sorting. Cayley Maze is universal: every finite deterministic sparse MDP is an MDP of a certain instance of Cayley Maze. We demonstrate how Cayley Maze enables control over complexity, simplification, and combination of its instances.

## 1 Introduction

Designing agents capable of generalizing across diverse tasks and environments is both a challenging and exciting problem in modern reinforcement learning. Open-ended learning, unsupervised environment design (UED), and curriculum learning remain attractive approaches to reach this goal (Hughes et al., 2024)

There has been a lot of progress in designing and implementing Open-Ended environments. For example, Genie (Bruce et al., 2024) or Craftax (Matthews et al., 2024). While the Genie is a huge achievement, we doubt that such an environment allows for producing challenging, algorithmic, or intelligent problems. On the other hand, some of the Unsupervised Environment Design(UED) algorithms(Beukman et al., 2024), (Parker-Holder et al., 2023) are still being evaluated on Minigrid (Boisvert et al., 2018) environments. In such an experiment-oriented field like Machine Learning, algorithms cannot be better than their evaluation procedures. We pose the question: what is a good evaluation procedure for Open Ended Learning? The partial answer is that variable complexity is necessarily its component. The core assumption of curriculum learning is the ability to produce observations that gradually become more complex. The diversity of parametrizable environments is an assumption of UED. Hence, it also relies on some notion of similarity/metric. This paper proposes a formal definition of the complexity of Reinforcement Learning environments. We discuss why the current heuristics, like state or action space cardinality, might be a naive estimate for this goal. We do it by translating certain concepts from the Language of Algebra, Deterministic Finite Automata theory, and Topology, and hopefully prove its usefulness. More than that, we introduce a novel Reinforcement Learning environment called Cayley Maze, which is, in a certain way, universal. We show that many important problems, such as sorting or Rubik's Cube, are specific instances of Cayley Maze.

## 2 Preliminaries

Here, we briefly recap some definitions of the algebraic approach to Automata theory. For more context we advise to read Pin (2021). *Monoid* is a set $M$ equipped with the operation $(\cdot) : M \times M \to M$ for which the following hold:

- Identity element: there exists $e \in M$ such that for every $m \in M$ $e \cdot m = m \cdot e = m$. The identity element is always unique.
- Associativity: for every elements $f, g, h \in M$, $(f \cdot g) \cdot h = f \cdot (g \cdot h)$

A *group* is a monoid whose every element has an inverse, i.e. for every $g \in G$ there exists $g^{-1} \in G$ such that $g \cdot g^{-1} = g^{-1} \cdot g = e$.

Given a group $G$, a *G-action* on the set $S$, is the map $\rho : G \times S \to S$, for which $\forall s \in S$ $\rho(e, s) = s$, and action commutes with multiplication: $\rho(g, \rho(h, s)) = \rho(g \cdot h, s)$ for all $g, h \in G$ and $s \in S$. A *homomorphism* between monoids $M$ and $N$ is a map between its sets $\phi : M \to N$, such that for every $a, b \in M$ $\phi(a \cdot b) = \phi(a) \cdot \phi(b)$. An *isomorphism* is a bijective homomorphism.

By $[n]$ we denote an $n$-element set $[n] = \{1, 2, \ldots n\}$. Given set $M$ and $S$, the set $M^S$ is a set of all functions from $S$ to $M$. A monoid of functions from $n$-element set to itself with the operation of composition is denoted by $End([n]) = [n]^{[n]} = \{f : f : [n] \to [n]\}$. The following theorem is the core idea of our environment:

**Theorem 1** (Cayley's theorem). Every finite monoid $N$ is isomorphic to some submonoid (subset) of $End([n])$ for certain $n \in \mathbb{N}$.

A subset $G \subseteq M$ of monoid $M$ is called a set of *generators* if it generates $M$, i.e. for every $m \in M$ there is a sequence $g_1, \ldots g_n$ such that $m = g_n \cdot g_{n-1} \cdot \ldots \cdot g_1$. A free monoid on the set $A$ is a set $M = A^*$, containing all finite sequences of elements of $A$, with the operation of concatenation, namely for $x = x_m x_{m-1} \ldots x_1$, $y = y_n y_{n-1} \ldots y_1$, $x \cdot y = x_m x_{m-1} \ldots x_1 y_n y_{n-1} \ldots y_1$.

A *congruence* on monoid $M$ is an equivalence relation $\sim$ on a set $M$, which is compatible with its operation: for every $a, b, c, d \in M$, $a \sim c$, $b \sim d \implies (a \cdot b) \sim (c \cdot d)$. Every congruence induces monoid structure on the set $M/\sim$ of equivalence classes on $M$ and a canonical homomorphism: $\pi : M \to M/\sim$, $\pi(a \cdot b) = [a \cdot b]_\sim = [a]_\sim \cdot [b]_\sim$. $M/\sim$ is called a *quotient* monoid of $M$. Given a monoid $M$ and its subset $L$, *syntactic congruence* on $M$ is defined as $a \sim_L b$ if for all $x, y \in M$ $xay \in L \iff xby \in L$. The quotient of this equivalence relation is called a *syntactic monoid*.

We call *Markov Decision Process* a tuple $(A, S, s_0, R, T)$ where $A$ is the set of actions, $S$ - the set of states, $s_0$ - initial state, $R : A \times S \to \mathbb{R}$ - reward function, and $T : A \to (S \to Pr(S))$ transition function, assigning the transition kernel $T(a) = T_a$ on $S$ to every state, where $Pr(S)$ stands for the space of probability distributions on the set $S$. We treat $T_a$ as a matrix of size $|S| \times |S|$, whose value at the row $s_2$ and column $s_1$ is denoted by $T_a(s_1)(s_2)$ meaning the probability of getting from $s_1$ to $s_2$ by the action $a$. If $T_a(s)$ are $0 - 1$ valued for all $a \in A$, $s \in S$, $T_a$ becomes a function $S \to S$, and we write $T_a(s_1) = s_2$ instead of $T_a(s_1)(s_2) = 1$.

For the mathematical convenience, we assume that the action set $A$ necessarily contains neutral (do nothing) action $e$. Then, its transition kernel $T_e$ is the identity matrix $id_S$. Markov Decision Process $(A, S, s_0, R, T)$ is called *sparse*, if there exists a a set of final states $F \subseteq S$, such that

$$\begin{cases} R(a, s) = 1 & T_a(s)(f) = 1 \text{ for some } f \in F \\ R(a, s) = 0 & \text{otherwise} \end{cases}$$

For given MDP with action space $A$, *trajectory* is a sequence of actions $\alpha = a_n \ldots a_1$; alternatively, it's an element of free monoid on $A$. Given a trajectory $\alpha$ we call its realization $T_\alpha = T_{a_n} \cdot T_{a_{n-1}} \cdot \ldots \cdot T_{a_1}$. Given the reward function $R : A \times S \to S$, we define $R : A^* \to Pr(\mathbb{R})$ to be the cumulative reward after moving along the trajectory $\alpha \in A^*$ from the initial state $s_0$.

A *transition monoid* of MDP $(A, S, s_0, R, T)$ is a set of trajectory realizations $M(T) = \{T_\alpha : \alpha \in A^*\}$, equipped with the operation of matrix multiplication. We say, that $R$ can be *factorized through* $M(T)$, if there exists $R' : M(T) \to Pr(\mathbb{R})$, such that for all $\alpha \in A^*$, $R(\alpha) = R'(T_\alpha)$. For brevity, we'll write $R$ instead of $R'$.

*Deterministic finite automaton* (DFA) is a tuple $(Q, A, T, I, F)$, where $Q$ is a set of states, $A$ - set of actions, $T : A \to (S \to S)$ - transition function (by $T_a$ we'll denote a transition kernel $T(a) : S \to S$), $I$ - set of initial states, $F$ - set of final states. Every DFA induces a directed graph on its states. Given DFA, its *transition monoid* is a set of matrices $\{T_\alpha : \alpha \in A^*\}$, equipped with the operation of matrix multiplication and identity matrix $T_e$ as a neutral element.

## 3 COMPLEXITY OF MARKOV DECISION PROCESSES

Loosely speaking, we say that two MDPs have the same complexity if their cumulative rewards are equal on every trajectory. We begin by proposing the extension of the notion of syntactic monoid

for general structures, such as functions, returning random variables. If one thinks that the condition $R(a) = R(b)$ as random variables is too strict, then one could compare expectations or replace it with some approximation, for example, by $d(R(a), R(b)) \leq \varepsilon$ after choosing certain metric on $Pr(\mathbb{R})$ and $\varepsilon \geq 0$. We'll assume that the reward function of every MDP can be factorized through its transition monoid, particularly it's true for every sparse MDP.

**Definition 1.** A reward congruence $\sim_R$ on MDP $(A, S, s_0, R, T)$ is a congruence on its transition monoid $\{T_\alpha : \alpha \in A^*\}$, such that for every $a, b \in M(T)$, $a \sim_R b$ if and only if for all $x, y \in M(T)$ $R(xay) = R(xby)$. Then, an *irreducible monoid* $M(R)$ of MDP is the quotient by the congruence relation $\sim_R$. R is well-defined on $M(R)$.

**Proposition 1.** A reward congruence $\sim_R$ on $M(T)$ for MDP $(A, S, s_0, R, T)$ is maximal among all congruences of the type: $\forall a, b \in M(T)\ a \sim b \implies R(a) = R(b)$.

*Proof.* For a congruence $\sim$ on $M(T)$ and some elements $a, b \in M(T)$, $a \sim b \implies \forall x, y \in M(T)\ xay \sim xby$. Hence $R(xay) = R(xby)$, and $a \sim_R b$. $\square$

There are multiple ways to define MDP's complexity: for example, one could measure the size of state space or action space. While these definitions are reasonable, the definition we propose captures a different kind of information.

**Definition 2.** Two MDP's $(A, S, s_0, R, T)$, $(A', S', s'_0, R', T')$ are equivalent if there is an isomorphism $\phi$ between their irreducible monoids $M(R)$, $M(R')$, preserving reward structure, i.e. $\forall a \in M(R), R'(\phi(a)) = R(a)$.

The definition 2 is equivalent to another one:

**Definition 3.** Two MDP's $(A, S, s_0, R, T)$, $(A', S', s'_0, R', T')$ are *equivalent* if there is a surjective homomorphism $\phi$ from $A^*\ A'^*$ or vice versa, such that reward structure is preserved: $\forall a \in A^*, R'(\phi(a)) = R(a)$. Hence, For deterministic MDPs with sparse binary rewards, the trajectory $\alpha \in A^*$ solves the first MDP if and only if $\phi(\alpha)$ solves the second MDP.

**Definition 4.** Order complexity of $MDP\ (A, S, s_0, R, T)$ is the cardinality of its irreducible monoid $M(R)$.

**Example 1.** Suppose we want to get on the right side of the grid, which has a width of 3 and infinite length. In other words, we are given an deterministic MDP $(A, S, s_0, R, T)$, where:

- State space $S = \mathbb{Z}_3 \times \mathbb{Z}$

- Initial state $s_0 = (0, 0)$

- Action space $A = \{(1, 0), (-1, 0), (0, 1), (0, -1)\}$

- Transition kernel $T(i, j)(a, b) = ((a + i) \mod 3,\ b + j)$

- Reward $\begin{cases} R((i, j)(a, b)) = 1 & (i + a) \equiv 2 \mod 3 \\ R((i, j)(a, b)) = 0 & \text{otherwise} \end{cases}$

By the definition 3, we define a reward congruence on $M(T)$:

$$(a, b) \sim_R (c, d) \iff \forall (x, y), (uv),\ R((x, y) + (a, b) + (u, v)) = R((x, y) + (c, d) + (u, v))$$

It is true if and only if $(x + a + u) \mod 3 = (x + c + u) \mod 3$, and so $a = c$, since $a, c \in \{0, 1, 2\}$. Hence the relation $\sim_R$ will have only 3 equivalence classes: $\{\{(i, j) : j \in \mathbb{N}\} : i \in \mathbb{Z}_3\}$, and the irreducible monoid will have only 3 elements. Hence, the irreducible monoid is isomorphic to $\mathbb{Z}_3$, and its order complexity is 3. The main conclusion is that MDP with infinite states and bigger action space might be equivalent (reduced) to MDP with three states and only one action.

We'd like to point out that even though deterministic sparse MDPs are very similar to DFAs, there exists a difference: while in RL, the episode stops if the agent reaches the final state, for DFA, the word belonging to its language might have an extension not belonging to it. Such difference can be eliminated by modifying the automaton or adding the termination action to the agent's action space. This modification looks useful and reasonable: without it, any finite MDP with a connected directed graph can be solved by the exhaustive search without any use of the environment's output.

# 4   CAYLEY MAZE

We propose a new Open-Ended Reinforcement Learning Environment: Cayley Maze. The agent's goal is to find the path between the initial and final vertices of a directed graph by choosing the edges to move along.

**Definition 5.** An instance of *Cayley Maze* is defined by the tuple $(m, n, T, i, F)$, where

- $m \in \mathbb{N}$ is the size of the action space , so $A = [m]$

- $n \in \mathbb{N}$ is the size of the state space, so $S = [n]$

- $T : A \to End([n])$ is the correspondence between action space and monoid generators; $G = T(A)$ is called the set of generators, and each generator is denoted by $T_a = T(a)$. For the function $T$ extended to $A^*$: $T(\alpha) = T_\alpha = T(\alpha_1 \cdot \alpha_2 \ldots \cdot \alpha_k) = T_{\alpha_1} \circ T_{\alpha_2} \circ \ldots T_{\alpha_k}$, the transition monoid of $T$ is the image $M(T) = T(A^*)$.

- $i \in [n]$ is an initial state

- $F \subseteq [n]$ is a set of final states

Then the instance induces a sparse deterministic MDP $([m], [n], i, R, T)$, where $R$ is

$$\begin{cases} R(a, s) = 1 & T_a(s)(f) = 1 \text{ for some } f \in F \\ R(a, s) = 0 & \text{otherwise} \end{cases}$$

The opposite appears also to be true:

**Proposition 2.** Every deterministic sparse finite MDP is an MDP of certain instance of Cayley Maze.

*Proof.* Since $(A, S, s_0, R, T)$ is deterministic, $T$ can be seen as a function $A \to (S \to S)$. Since MDP is sparse, $R$ is completely defined by the subset of final states $F \subseteq S$. Then, after enumerating $A$ and $S$, $(|A|, |S|, T, s_0, F)$ is an instance of Cayley Maze with the same MDP. $\square$

Cayley Mazes are parametrizable by the size of the state space $n$, the size of the action space $m$, monoid generators $T(A)$, and the choice of initial and final states. From now on, all discussed Reinforcement Learning environments and their MDPs are assumed to be deterministic and sparse. While the transition between MDP and Cayley Maze formalisms is tautological, we see it valuable for several reasons.

## 4.1   CAYLEY MAZE IS NATURAL

Cayley Maze naturally generalizes many important problems. Such problems deserve a special name:

**Definition 6.** An instance of the Cayley Maze is *natural* if the following holds:

$$(\forall \alpha, \beta \in A^* \ \exists i \le n \ T_\alpha(i) = T_\beta(i)) \implies T_\alpha = T_\beta$$

Restating this property: if two trajectory realizations are equal at some state $i$, then they are equal at any other state. Such remarkable property can be used for evaluating agent's generalization capabilities and architecture's inductive biases.

The problem of sorting the array of length $n$ can be seen as the instance of natural Cayley Maze, where:

- state space is the group of all $n$-element permutations $S_n$,

- action space - some subset of $S_n$, generating it. For example, it could be the set of all transpositions $\{(i, j) : i, j \le n\}$.

- the transition monoid is defined by the left multiplication of the state by the action

- the initial state is the unsorted number array seen as a permutation, and the final state is the identity permutation. In this case every winning trajectory $\alpha \in A^*$ gives the right order $T_\alpha$

Rubik's Cube is another such problem. Enumerating the squares of Rubik's Cube allows to translate the problem just like in the case of sorting; the only difference is that actions will have different kinds of permutations.

## 4.2 CAYLEY MAZE HAS VARIABLE COMPUTATIONAL COMPLEXITY

Various subfamilies of the Cayley Maze have different computational complexity: it has been shown that the problems of sorting the array and solving the Rubik's Cube can both be represented as instances of the Cayley Maze. The sorting problem has polynomial time complexity, while the problem of finding the optimal solution of Rubik's Cube is NP-Complete (Demaine et al., 2018).

## 4.3 MOST OF THE REINFORCEMENT LEARNING ENVIRONMENTS ARE NOT UNIVERSAL

Some of the most popular environments used to evaluate UED algorithms, like Minigrid mazes (Boisvert et al., 2018), make heavy use of the underlying geometric structure of its state space. We note that any Open-Ended environment, which can be solely represented by moving on the 2-dimensional grid (i.e., for which the directed graph of its MDP can be embedded into the plane respecting grid structure), is not universal in the sense of proposition 2: for example an MDP with three states $\{A, B, C\}$ and one action $a$: $A \xrightarrow{a} B \xrightarrow{a} C \xrightarrow{a} A$ cannot be embedded into the plane, since otherwise agent would have to always move in the same direction and return to the initial state. What is less obvious is that such environments are not universal even in the sense of definition 2:

**Proposition 3.** There exists an MDP that is not equivalent in the sense of definition 2 to any MDP whose directed graph is planar. Consequently, such MDP cannot be represented by moving on a 2-dimensional grid.

The proof of this fact is due to Book & Chandra (1976). The witnessing automaton has only 7 states and 6 actions. The further development of this topic and the applications of topology for measuring the complexity of finite automata can be found in Bonfarte & Deloup (2018)

## 4.4 MODIFICATION AND SIMPLIFICATION OF EXISTING INSTANCES

Given an abstract MDP or the set of game instructions, it is often unclear which modification would make it simpler or harder. But it's certainly possible for Cayley Mazes: painting all Rubik's cube faces into black makes it much easier to solve. In other words, given the MDP $M$ whose transition monoid acts on the set $S$, and the coloring of $S$ - a surjective function $h : S \to K$, it's not hard to build *quotient* of $M$ by $h$, whose construction is similar to its group-theoretic analog.
Many constructions on groups, such as products, allow to efficiently combine existing MDPs to produce new ones.

## 5 APPLICATIONS AND IMPLEMENTATION DETAILS

We implement Cayley Maze as a parametrizable environment in JAX with a Gym interface. It allows the sampling of any MDP with predefined action space and state spaces or the modification of the existing transition monoid, initial and target states.
Cayley Maze may be used in the evaluation procedure of every learning problem, which has a sequential structure and where generalization or emergent complexity of observations are crucial. Some of the proposed scenarios are:

1. Since universality is essential for UED, Cayley Maze may be used for the evaluation of UED algorithms

2. It is possible to create environment samplers whose instances have a common structure. For example, it might be MDPs whose transition monoids are simple groups or environments that represent $n \times n \times n$ Rubik's Cube.

3. It is possible to evaluate not only on the subfamilies of environments but also on various combinations of these environments. For example, we can check whether the agent, which performed well on some instances, would perform well on its product.

4. Another way to test generalization capabilities is to use the local property of natural Cayley Mazes, as in 4.1. For example, evaluate the agent's performance on the initial states that were unreachable during training.

5. Since the process of creating a new MDP can be seen as an MDP, it can be expressed as an instance of Cayley Maze. Hence, the UED scenarios where the teacher learns to build an environment without student become possible.

Another interesting feature of the implementation is that while constructing the most general reinforcement learning environment, one might expect the explosion of the teacher's action space. The implementation allows customization of the desired trade-off between the size of action space, representation dimension, and the power of the edit per step.

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
