# OpenReview forum: "Cayley Maze: Universal Open-Ended Reinforcement Learning Environment"
_ICLR.cc/2025/Conference — Submitted to ICLR 2025_

### Official Review · Reviewer_5aVY · 2024-10-23

**Soundness:** 1
**Presentation:** 1
**Contribution:** 1
**Rating:** 1
**Confidence:** 2

**Summary:**

The authors introduce Cayley Maze, a universal mathematical framework based on Finite Automata and Group Theory machinery, that may be used to represent an open-ended reinforcement learning environment that naturally generalizes problems like solving Rubik’s Cube, sorting, and integer factorization, and that enables control over complexity, simplification, and combination of its instances.

I found this paper extremely unclear about its goals and its main messages, and I’m strongly in favor of rejecting it.

**Strengths:**

No idea, sorry

**Weaknesses:**

- the authors claim that Cayley Maze is a novel open-ended reinforcement learning environment, but they never define what an open-ended reinforcement learning environment is.
- the authors mention what it is possible to do with the Cayley Maze, rather than what they did
- the paper does not have a conclusion
- the paper is very poorly written, with many unclear statements, english mistakes, typos, unclear figures with poor captions, etc.
- no related work section, no comparison to anything, no ablations...

**Questions:**

I’m not using this section only to ask questions, but also to criticize the current form and to suggest improvements to the authors.
- line 39: “we doubt that such an environment allows for producing challenging,” What makes the authors doubt so? Did the authors perform experiments to backup their doubts? Science is not about expressing raw beliefs and doubts, it is about establishing facts.
- line 43: “algorithms cannot be better than their evaluation procedures” → how can the authors compare an algorithm to an evaluation procedure? Don’t the authors mean “algorithms cannot be better than what their evaluation procedures allows”? What do the authors mean exactly?
- line 46: “The diversity of parametrizable environments is an assumption of UED, hence it also relies on some notion of similarity/metric.” I don’t understand the sentence. Can the authors be more explicit about the relation between the diversity of parametrizable environments and the notion of similarity/metric?
-line 50: “We … hopefully prove its usefulness”. → I’m afraid I’m not convinced by the “proof”.
- why is Theorem 1 useful in the context of the paper? What claim about open-ended RL does it support?
- line 107: “Loosely speaking, we define two MDP’s to have same complexity, if the corresponding trajectories yield the same cumulative reward.” → How do the authors define the fact that 2 trajectories correspond to each other?
- line 138 (Example 1): “on the right side of the grid” → what grid? We are missing some context here…
- line 172: what is a monoid generator?
- line 217: “is simple in group-theoretic sense” what does this mean?
- line 250: “whose construction is similar to it’s group-theoretic analogue.” → why “similar”? What is the point?
- all figures should be self-contained: the caption should specify what is the environment, what the figure shows, what we should conclude from watching it. Figure 1 does not have labels on the x and y axis, we don”t even know what it measures. This is not acceptable. Do the figures show a single realization or a mean? What about the variability of the results? These mistakes are enough to kill the paper, even if the paper had a clear message.
- same for figures 2, 3 and 4
-the paper should come with a conclusion
# Typos, minor errors:
- there should alsways be a space before “(“, but none in “) ,”
- on the figure X → in Figure X (many times)
- the definition X → Definition X. Same with proposition.
- Since MDP → Since an MDP. A lot of articles are missing
- Note, that → Note that (many times)
- MDP’s → MDPs (many times)
- line 213: in this cases → these cases (or this case)
- it’s can be either “it is” (do not use contractions in scientific papers, no “doesn’t” etc.) or its (the possessive). Please correct them all.
- line 226: (Demaine E., 2018). → is it rather “Eisenstat& Demaine”? The reference “Eisenstat S. et al. Demaine E.” looks incorrect. Use bibtex to get proper references.
- Fig 3: of of

**Details Of Ethics Concerns:**

Mathematics are harmless (I hope)

---

> ### Author Response · Authors · 2024-12-04
>
> We find your review highly unconstructive. We’d like to discuss some issues you’ve mentioned:
>
> >the authors claim that Cayley Maze is a novel open-ended reinforcement learning environment, but they never define what an open-ended reinforcement learning environment is.
>
> By Open-Ended reinforcement learning environment we mean a parameterized family of MDPs.
>
> > The authors mention what it is possible to do with the Cayley Maze, rather than what they did.
>
> As written in the abstract, the goal of the paper was to “demonstrate how Cayley Maze enables control over complexity, simplification, and combination of its instances”, and that’s what we did.
>
> > the paper is very poorly written, with many unclear statements, english mistakes, typos, unclear figures with poor captions, etc.
>
> We’ve corrected all the typos you’ve mentioned and many others.
>
> > no related work section, no comparison to anything, no ablations...
>
> To the best of our knowledge, there do not exist other universal RL environments (the universality of OMNI-EPIC is not proved or guaranteed), or non-trivial definitions of the complexity of MDPs.
>
> > line 39: “we doubt that such an environment allows for producing challenging,” What makes the authors doubt so? Did the authors perform experiments to backup their doubts? Science is not about expressing raw beliefs and doubts, it is about establishing facts.
>
> Our conclusion about Genie is based on the examples of generated environments from the article - most of them are platformers, which by any means are not conceptually challenging.
>
> > line 46: “The diversity of parametrizable environments is an assumption of UED, hence it also relies on some notion of similarity/metric.” I don’t understand the sentence. Can the authors be more explicit about the relation between the diversity of parametrizable environments and the notion of similarity/metric?
>
> Saying that the set of objects is diverse by definition means that the objects differ from each other. Two objects are similar if they do not differ. Metric is the concept which allows to measure similarity: two objects are equal if the distance between them is 0.
>
> > why is Theorem 1 useful in the context of the paper? What claim about open-ended RL does it support?
>
> It shows that every finite monoid (and hence every finite MDP) can be ”encoded”. Without that universality statement would be wrong.
>
> > line 107: “Loosely speaking, we define two MDP’s to have same complexity, if the corresponding trajectories yield the same cumulative reward.” → How do the authors define the fact that 2 trajectories correspond to each other?
>
> Corrected: two MDPs have the same complexity if their cumulative rewards are equal on every trajectory.
>
> > line 138 (Example 1): “on the right side of the grid” → what grid? We are missing some context here…
>
> The state space in the example is an infinite grid. Look at the lines 139-140.
>
> > line 172: what is a monoid generator?
>
> Look at the line 70.
>
> > line 217: “is simple in group-theoretic sense” what does this mean?
>
> https://en.wikipedia.org/wiki/Simple_group
>
> > line 250: “whose construction is similar to its group-theoretic analogue.” → why “similar”? What is the point?
>
> By that we mean that the construction of the quotient by the function is similar to the quotient by the subgroup.

---

### Official Review · Reviewer_EVgt · 2024-10-30

**Soundness:** 1
**Presentation:** 2
**Contribution:** 2
**Rating:** 3
**Confidence:** 3

**Summary:**

This paper proposes a definition of complexity for deterministic MDPs; it also introduces Cayley Maze, which can model a general class of deterministic MDP problems for unsupervised environment design (UED) and other frameworks with an extra level of complexity over the traditional reinforcement learning framework. The paper first views the sequences generated in an MDP from the perspective of Finite Automata theory and proposes the new complexity accordingly. Further, the paper introduces the Cayley Maze and shows that every deterministic finite MDP with sparse rewards corresponds to an instance of the Cayley Maze. Limited experiments are performed to examine the performance of different UED algorithms on some Cayley Maze instances.

**Strengths:**

The strengths of the paper are its originality and potential significance:
1. The perspective developed in this paper is novel to the knowledge of the reviewer. The proposed complexity measure seems grounded and removes redundancy in the MDP’s structure.
2. The paper has some potential to inspire and be useful to future research on UED and other fields like meta reinforcement learning. The proposed Cayley Maze seems to have the potential to be useful for these fields.

**Weaknesses:**

While the paper has notable strengths, there are several significant weaknesses that impact its overall contribution:
1. The quality of the paper is poor. The paper is rough and seems to be incomplete. Specifically,
* The experiments seem to be limited, and their conclusions are unclear. There are only three small experiments provided, and their implications are unclear. What are the goals of the experiments? Other than some results of the three algorithms’ performance in a few instances, what can the reader learn about the proposed environment? Are the current set of experiments sufficient? Having more detailed discussions in the experiment section and including a conclusion section will help clarify the implications. If needed, more experiments should be conducted to justify the utility of the proposed environment.
* Sufficient details about the experiments are not provided despite the fact that there are a few more pages available. It may be useful to address the following concerns:
  * What exactly are the used environment instances? Is there a way to describe them better? From reading the paper, I cannot understand what these environments are and their implications. Including some more details even in the appendix would help.
  * While there isn’t any discussion or citation for the domain randomization algorithm, the reader might not be able to figure out what it is.
  * What does the y-axis in all the plots represent? In addition, the x-axis should also have a label for clarity.
  * The performance variation across different random seeds is not provided.
  * It’s unclear what Figure 3 represents. The text also doesn’t give sufficient information.
* While the proposed Cayley Maze is general and includes both easily solvable and difficult instances, how to generate interesting instances is not thoroughly discussed apart from a very short discussion in Section 4.4. In other words, what makes Cayley Maze a good class of problems to work on? And how can it be used?
2. While a new complexity measure is proposed, the paper doesn’t provide any further theoretical results, empirical implications, or justifications for its usefulness.
3. The paper also needs to improve its clarity. In addition to the missing information mentioned above, in the Questions section, there is a list of small issues that reduce the paper’s clarity.

**Questions:**

Other than the questions asked in the Weaknesses section, here are questions that might affect the evaluation:
1. In Line 213, what does it mean by saying $M(T)$ acts on the state space $S$ by multiplication of action and state? And is it good for evaluating agents’ generalization capabilities and architecture’s inductive biases?

Other minor questions or suggestions:
* Line 021 - Inaccurate claim in the abstract: Only deterministic sparse MDPs are shown to be instances of Cayley Maze.
* Line 060 - Confusing expression: It’s better to put a comma after $m\in M$.
* Line 066 - Undefined operation: What does $g\times h$ mean? Is it the same as $g\cdot h$?
* Line 078 - Delayed definition: The definition of isomorphism should be before Theorem 1. Also, it would be good to add a citation that contains the proof of the theorem.
* Line 101 - Unclear acronym: Does DFA mean deterministic finite automaton? If so, be consistent with the definition in line 99.
* Line 117 - Confusing expression: Having the definition for congruences that preserve reward structure after the proposition statement confuses me. It might be better to remove it.
* Line 158 - Unclear concepts: What are words and language here?
* Line 173 - Unclear notation: What is $Im(T)$?
* Line 247: The sentence “But it’s certainly possible for MDP’s” does not connect to the previous sentence and is confusing. Do you mean “for some MDPs”?

---

> ### Author Response · Authors · 2024-12-03
>
> Thank you for the review. We would like to comment on several issues you've mentioned:
>
> > While the proposed Cayley Maze is general and includes both easily solvable and difficult instances, how to generate interesting instances is not thoroughly discussed apart from a very short discussion in Section 4.4. In other words, what makes Cayley Maze a good class of problems to work on? And how can it be used?
>
> Firstly, every finite group corresponds to the unique natural Cayley Maze, forming a large class of predifined environments. Since much more is known about groups than about RL environments, this framework provides the control over the complexity and variety of environments by using group theory techniques.
>
> Secondly, since all RL environments are created by humans, they inherit human biases, such as the geometry of two or three-dimensional space (see section 4.3). Cayley Mazes do not have this problem.
>
> Thirdly, the rich symmetric structure of natural Cayley Mazes (see definition 6) which can be utilized by the agent makes every such problem interesting.
>
> > While a new complexity measure is proposed, the paper doesn’t provide any further theoretical results, empirical implications, or justifications for its usefulness.
>
> Could you answer what is the usefulness of the definition of convergence in distribution, or of the Bayesian Information Criterion? Definitions are created in order to capture certain properties of the objects of these definitions. Our definition encapsulates the notion of irreducible (lower-bound) complexity. Informal but accurate intuition: complexity of an MDP is the minimal amount of space required to store all the winning trajectories of this MDP.
>
> > The paper also needs to improve its clarity. In addition to the missing information mentioned above, in the Questions section, there is a list of small issues that reduce the paper’s clarity.
>
> All the issues from the questions (except line 60) are fixed in the updated version.
>
> > Line 158 - Unclear concepts: What are words and language here?
>
> Language of automaton is the biggest subset of the free monoid which is accepted by this automaton. The word is an element of the free monoid on which the automaton operates. For more information please read J. Pin from the references section.

---

### Official Review · Reviewer_J5De · 2024-11-04

**Soundness:** 2
**Presentation:** 2
**Contribution:** 4
**Rating:** 5
**Confidence:** 5

**Summary:**

This paper proposes both a spaces of parameterised environments, and a complexity measure for these environments. The environments are effectively problems involving the navigation of Cayley diagrams through the generators of a monoid, and the complexity of an environment being the size of the smallest monoid the environment has a reward-preserving homomorphism to. This space of environments, dubbed Cayley Mazes, are implemented in JAX, and they are tested as benchmarks for UED algorithms.

**Strengths:**

This paper addresses a real need in the field of Open-Ended Learning and UED. Moreover, it does so in a very natural and powerful way, brining to bear a powerful suite of existing tools from abstract algebra. The idea that environments that are more symmetric are simpler all else being equal is intuitive and natural. Moreover, the structure of monoids gives ample opportunity to see emergent structure from our environment design algorithms if our algorithms are capable of effectively searching for this structure. It is clear to me that this is a promising direction for the field of UED and that the environment in JAX is a strong contribution.

**Weaknesses:**

Unfortunately, I believe there are some serious weaknesses with the current form of this paper which will make it difficult for it to achieve that impact. My two main concerns are:
1) The exposition is likely to loose most readers in UED
2) The empirical evaluation and presentation does not meet the standards in RL and UED, so it is difficult to use as a baseline.

I believe both of these issues are resolvable, though likely not in the rebuttal period. I would encourage the authors to try to really over-do it on fixing points 1 and 2. I believe many in the UED space would be excited about this work if those points were soundly addressed. Unfortunately, due to these concerns I will currently have to recommend rejection. More specific details of these concerns are described below.



### Improving the Exposition

Much of the paper is too fast for people without a strong abstract algebra background, which the vast majority of the target audience does not have. I think the paper will effectively have to teach the readers enough of abstract algebra to get through the main ideas of the paper, and do so assuming they have never seen the field before. I believe this is quite doable, but will require a good amount of effort.


The paper is quite fast through pages 2 and 3. At the same time, it is essential that the reader understands these ideas deeply as the intuition being drawn on here relies on a quite robust understanding of the monoids. The paper needs to impart the intuition that they have something to do with complexity and symmetry that only becomes obvious after the reader really groks the concept. This will require some hand-holding -- "alternatively its an element of the free monoid on A" is quite a complicated throw-away phrase for someone who just was told about free monoids, and most readers would not realise that it is not saying anything complicated. It would be better to introduce this idea in a few lines, with an example so they understand why that framing makes it easier to think later.

I think that if example 1 was introduced much earlier, ideally in the introduction, and used to introduce all of the definitions, it would be much easier to follow.  Moreover, I would include a lot of visuals, namely Cayley diagrams would make everything much easier to follow and would make the ideas concrete.

In definition 5, it's unclear how exactly this maps on the UED, in that the parameters of the UMPD being proposed aren't defined explicitly. I would suggest defining the Cayley Maze concretely as a UMPD to avoid this confusion. Without this, it is hard to know what the scope is of a specific environment, and it is then hard to interpret the experiments.

Proposition 2 is a bit narrower than is claimed in the intro (constraining to deterministic sparse and finite), you should mention these constraints in the intro when you mention these results.

In section 4.3, you should reference OMNI-EPIC as it is also universal, and comes with a notion of complexity (program length).

There are often references to group-theoretic concepts that the reader doesn't know, but would need to understand to understand the rest of the paper.  For instance, the reference to "the symmetries of the 3 cube", "Group of order 6" and "wreath product" in the experiments section all will be lost on readers.


**Some minor clarity concerns:**
 - 016 should be "A" formal definition of complexity
 - 038 "For" should not be capitalised
- 085 in the definition of syntactic congruence, the right hand side of the implication should be xby \in L not xby \in M
 - 088 It's not clear what is meant by a transition kernel
 - 095 It's not clear what is meant by "it's realisation" or "a resulting kernel"
 - 100 Referencing this as matrix multiplication is confusing since T_\alpha are functions, not matrices. I think this means composition where the "matrix" being referenced is the probability transition matrix representation of T, but that is a few more levels of indirection than necessary. Saying composition here is fine (relying on the implicit fact that a function of a random variable also gives a random variable)
- 107 it's not clear what the "corresponding trajectories" would be
- 317 there is a stray "i"


### Improving the Evaluation
The goal of the evaluation section ought to be to set this up as a benchmark that others in UED can aim to improve on. As it stands, it is missing several components for that.  Namely:

- The evaluation protocol should match that in prior UED works
- The task should be clearly delineated in a easy-to-hard order with the easiest ones being solved and the hardest ones only partially working
- There should be a clear ideas what the scores mean in each setting

*Evaluation protocol:*
 It should follow the conventions of evaluation in papers like PAIRED and PLR, describing an explicit test set of levels and corresponding bar charts, the inter-quartile scores as described in [1]. These test levels should vary from easier examples to hard examples and should be diverse while giving signal to partially-working methods. There should also be visual depictions of the levels being generated, since the level-generation is what methods would be evaluating.  The return plots should all have the appropriate error bars. The goal for being rigorous on this is to lay the foundation for researchers who use this environment as a benchmark to have the information they need to evaluate their approaches, and for the readers to have the information they need to tell if this is a benchmark that is too hard, too easy, or measuring current frontier progress. It should be clear that the methods work on some of the environments (for instance the Cayley Mazes equivalent to minigrid), and fail on others (the full Rubix cube setting). Ideally, there would be some instances of nearly-solved Rubix cubes in the test set that the methods show some performance on so others have a signal to hill-climb on.

In Figure 3 it looks like PLR is still improving, It would be best to run the methods for longer to make sure the baseline reports the best numbers for current methods so that they can be more readily beaten with new methods. It also looks like all of the methods need to be tuned, or at least it is difficult to tell if they have been tuned will given the information provided. This is also why it would be best to give some easy tasks, or easy instances of the given tasks to show the methods are tuned enough to work a bit. Steps per second should also be reported for the environment during training, since this is a critical limiting factor and selling point for any Jax environment.

*Clear Task Definitions:*
Given the wide space of Cayley Mazes, it would be best to delimitate specific subsets which can serve as self-contained environments to test UED aparoches on. This is like how minigrid has a range of  different settings provided by default, along with current algorithms performances on these settings. These tasks should be clearly delineated and named so that other papers could easily reference the definition of the task that they are training on.  Moreover, it is best that this paper as the benchmark defines several of these tasks since it could then be presented as standard suites, and authors would no longer be able to easily pick the problem that their method performs well on.

*Clear meaning of tasks:*
It would be best of there was some sort of visualisation of the tasks, along with a much slower non-group-theoretic description of them. The performance on these problems is only useful to the reader if they understand how to interpret the difficulty of the tasks. As such, the main message I think most would get from these are "these seem to be hard tasks", but they get no sense of if these are "proving a novel theorem" levels of hard, or something more manageable for a new algorithm.

[1] Agarwal, Rishabh, et al. "Deep reinforcement learning at the edge of the statistical precipice." Advances in neural information processing systems 34 (2021): 29304-29320.

**Questions:**

How do you think the edits in methods like ACCEL could best be applied to Cayley maze?

---

> ### Author Response · Authors · 2024-12-03
>
> We would like to thank you for the thorough review; we find it extremely useful. Comments on the several issues you've mentioned:
> > In definition 5, it's unclear how exactly this maps on the UED, in that the parameters of the UMPD being proposed aren't defined explicitly. I would suggest defining the Cayley Maze concretely as a UMPD to avoid this confusion. Without this, it is hard to know what the scope is of a specific environment, and it is then hard to interpret the experiments.
>
> > Proposition 2 is a bit narrower than is claimed in the intro (constraining to deterministic sparse and finite), you should mention these constraints in the intro when you mention these results.
>
> Fixed.
> > How do you think the edits in methods like ACCEL could best be applied to Cayley maze?
>
> Our implementation supports ACCEL: the initial and final states can be modified by moving one step in a random direction, and the transition function of the MDP can be modified by multiplying the transition monoid generators by some transposition.

---

### Official Review · Reviewer_nztC · 2024-11-04

**Soundness:** 3
**Presentation:** 2
**Contribution:** 2
**Rating:** 6
**Confidence:** 3

**Summary:**

This paper proposes an interesting RL environment that has theoretical properties and tuneability to control for complexity over different MDPs problems. However, rather than an environment or a family of environments, the proposed Cayley Maze is a tool to interpret a wide range of MDPs.  The environment is built from a mathematical foundation using group theory and finite automata concepts. Experiments combine the approach with  PAIRED (Protagonist Antagonist Induced Regret Environment Design) and PLR (Prioritized Level Replay) to provide evidence that  that Cayley Maze can support diverse unsupervised environment design approaches.

**Strengths:**

- The paper proposes a rigorous mathematical framework using algebraic structures, specifically group theory and finite automata theory, to define the complexity of Markov Decision Processes (MDPs). This mathematical rigor provides a foundation for measuring and controlling complexity more precisely, which is valuable in the field of reinforcement learning (RL), particularly for curriculum learning and adaptive agent development.
- Claim of universality: the Cayley Maze can represent any deterministic MDP.
- Problem representation through a formal method. This structured approach offers researchers a reliable way to analyse task properties and provides a clear foundation for designing new tasks within Cayley Maze. This can lead to better reproducibility and comparability of results, as tasks can be rigorously defined and consistently replicated across studies.
- Applicability to unsupervised environment design: this approach could be particularly useful to create specific benchmarks for RL evaluation that currently relies on benchmarks with unclear properties and complexity.

**Weaknesses:**

1. While the principles are interesting, it's unclear how this can be used on a large variety of benchmarks to experiment with open-endedness.  It would strengthen the work if the authors demonstrated how Cayley Maze aligns with or diverges from existing benchmarks in terms of complexity, adaptability, and the variety of problems it can handle
2. The paper discusses the variable complexity of Cayley Maze but could further demonstrate this aspect through empirical tests showing how increasing or decreasing complexity affects agent performance.
3. Can the authors discuss how can the approach be adapted to work with stochastic environments? Since many real-world applications involve stochastic elements, this is a crucial gap. Cayley Maze’s deterministic nature could limit its real-world applicability, as it does not address environments where randomness or partial observability plays a role. The paper could benefit from a discussion or proposed modifications for incorporating stochastic transitions, which are often essential for testing generalization and robustness in RL agents.
4. While I appreciate the foundation in group theory, monoids, and algebraic structures, this is also a limitation. For those that are not too familiar with the mathematical formalism, it would be good to read a more thorough discussion of a longer list of problems that can be formalized with Calyey Maze, and those that they cannot be formalized in such a way. To improve accessibility, the paper could offer more concrete examples of specific problems that Cayley Maze is well-suited to address and explicitly identify cases where Cayley Maze’s mathematical structures may not apply. This would clarify both the scope and limitations of the environment for a wider audience.
5. One question remains on whether the RL strategies can generalize. The paper would benefit from experiments or discussions that illustrate how agents trained in Cayley Maze fare on distinct yet related tasks. For example, are there cases where skills learned in one Cayley Maze configuration (like Rubik’s Cube) transfer effectively to another (e.g., different types of sorting problems)? Providing empirical evidence or a theoretical argument for generalization would significantly strengthen the paper’s claims.
6. I could not find the JAX implementation: this would be very useful for reproducibility.
7. The language is at time informal.

**Questions:**

I would like to see some modifications to the paper to address the weakness above. In particular, point 2, 3, 4 and 5 are quite central and addressing those point could strengthen the paper.

---

> ### Author Response · Authors · 2024-12-03
>
> Thank you for the review! We would like to comment on the several weaknesses you've mentioned.
>
> 3. The proposed complexity of MDPs works for MDPs (deterministic and stochastic) with factorizable reward function. The stochastic modification of the Cayley Maze is straightforward by replacing deterministic transition function with the stochastic one (or, in other words, $T: A \to End([n])$ becomes   $T: A \to ([n] \to Pr([n]))$ ).
>
> 4. Since Cayley Maze is universal, any environment with finite state space and binary episode outcome can be expressed as an instance of Cayley Maze. Finiteness condition can be omitted if we know how to represent infinite state space effectively. However, most of the environments are not natural Cayley Mazes.
>
> 5. The experiment 1 from the previous version of the paper is exactly about that: an agent was trained on two environments and evaluated on the wreath product of these environments, solving which required a non-trivial combination of skills acquired on each of them.

---

### Author Response · Authors · 2024-12-03

We would like to thank the reviewers. Since the experiments section deserves to be extended, we’ve removed it from the current version and left it for future work. We have fixed the typos; one more significant modification is the formal definition of the natural Cayley Maze (definition 6). Specifically, every Cayley Maze on the Cayley graph is natural.

---

### Meta-Review · Area_Chair_SCrB · 2024-12-19

**Metareview:**

This paper introduces Cayley Maze as a general open-ended reinforcement learning environment that encompasses well-known problems such as the Rubik's Cube, sorting, and factorization. Further, the authors argue that Cayley Mazes include every finite deterministic sparse MDP.

The framework introduced is novel and interesting, and is laid out with mathematical rigour.

However, the presentation was rather terse and rushed, which may make it difficult for most readers to follow. Further, the current revision is lacking any empirical evaluation. There were some experiments in the original submission but the authors have removed it from the latest revision and left it for future work.

As such, although this idea has merit, I do not believe it is ready for publication yet.

**Additional Comments On Reviewer Discussion:**

Most of the concerns were around clarity of exposition and implementation details. The discussion seems to have prompted the authors to completely remove the empirical evaluations. Some of the typos were fixed, but the clarity of exposition still needs to be improved.

---

### Decision · Program_Chairs · 2025-01-22

Reject